# Exploring and Developing a Scale Using Item Response Theory for Sport Psychological Skills in Speed Skaters

**DOI:** 10.3390/ijerph19138035

**Published:** 2022-06-30

**Authors:** Jung-Hoon Nam, Bong-Arm Choi, Eun-Hyung Cho

**Affiliations:** 1Department of Sports Healthcare, Catholic Kwandong University, Gangneung 25601, Korea; n7j7h782@gmail.com; 2Department of Physical Education, Daegu University, Daegu 38453, Korea; ba3665@naver.com; 3Department of Sport Science, Korea Institute of Sport Science, Seoul 01794, Korea

**Keywords:** sports psychological skill, speed skaters, item response theory, BSEM

## Abstract

The purpose of this study was to develop a sports psychological skills scale for speed skaters and examine the validity of the scale. In order to accomplish the purpose of this study, skaters from around the world were set as a population, and then data from 456 athletes were collected using convenience sampling from the athletes participating in the 2020–2022 World Championships and the Beijing Winter Olympics. For analysis, V coefficient, Parallel Analysis, Exploratory Structural Equation Modeling, Bayesian Structural Equation Modeling, Maximum Likelihood CFA, and analysis of Multi-Group Confirmatory Factor Analysis were carried out by using WINSTEPS 3.65 and MPLUS 7.04 programs. The level of statistical significance was all set at α = 0.05 and Exploratory Structural Equation Modeling, Bayesian Structural Equation Modeling, Maximum Likelihood CFA, and Multi-Group Confirmatory Factor Analysis fit used TLI, RMSEA, the concept of reliability. The findings of the study were as follows: First, the factorial structure of SPSS was extracted as five factors with 17 items. Second, the analysis of MCFA on the transformative leadership scale, according to gender differences, was carried out, and cross validity was fulfilled.

## 1. Introduction

Sports scientists have continuously delved into research to identify the answer to what determines athletic performance in a competitive setting. In recent years, researchers have employed detailed and empirical approaches, considering the characteristics of the sport of interest and athletes to identify the determinants of athletic performance [1,2].

Early studies on athletic performance have attempted to predict and describe the factors for performance primarily based on athletes’ physical fitness and skills. However, studies [1,2,3,4] show that athletic performance is determined by physical, skill-related, psychological, and physiological factors, and this view has been widely accepted. Hence, field models illustrating the relationship between athletes’ psychological resources and performance in sports settings [5,6] have been previously proposed, and several sports psychology studies have been conducted to identify the psychological factors predicting athletic performance.

Sports psychology suggests athletes’ psychological resources are the resources for emotional regulation determining athletic performance if the physical, technical, and physiological factors are relatively stable and invariant throughout a game [7].

During a game, athletes experience a spectrum of emotions, depending on their surrounding environment and game status, and negative emotions are particularly more prevalent than positive emotions [8,9]. For this reason, most athletes frequently attain poorer results during an actual game than during training. This is because negative emotions perceived during a competitive situation induce negative physiological, biochemical, and behavioral symptoms and, thus, adversely impact technical performance [10,11]. Therefore, effectively controlling and regulating athletes’ negative emotions is the core strategy to enhance athletic performance in the sports psychology field.

Typically, the ability to regulate emotions from various negative emotional stimuli during training and competitive sports settings is considered in the remit of sport psychological skills (SPS). SPS is defined as psychological regulatory skills that maximize performance by controlling, suppressing, or converting negative psychological factors, such as anxiety, fear, and frustration, perceived by athletes, which hinder performance, to positive psychological factors, such as confidence [12,13,14].

Today, applying SPS throughout sports settings is broadly divided into two categories: (a) psychological skills converting negative thoughts perceived during training or games into positive thoughts that boost performance, such as mental training, the progressive relaxation technique, attention control training, and self-confidence skills and (b) psychological skills, such as imagery and image training, enabling athletes to imagine the game and training settings in advance to identify the essential factors needed for training or preparation for games, so as to adapt to, control and regulate potential negative psychological situations. In recent years, SPS has been specialized in specific types of sports, training and game environments.

Notwithstanding this trend, studies on SPS regarding speed skaters [15,16,17,18,19] primarily present the relationship between the significance of SPS and athletic performance without empirically identifying SPS tailored to the features of skating and skaters [20]. These studies suggest that anxiety, depression, fighting spirit, confidence, goal setting, image, team harmony, and determination require SPS to enhance skating performance. Most studies have attempted to explain the relationship between skating performance and psychological skills using scales for a single aspect of affect, such as state of anxiety or trait anxiety, or a psychological skills inventory that was developed based on another sport.

Martens [21] suggested that using a sport-specific psychological skills scale, rather than a scale developed for universal use in several sports, is more reasonable when empirically measuring psychological factors. Hence, developing an SPS scale tailored to the skating sport and to skaters is essential to maximize skaters’ SPS and ensure optimal performance.

Studies that developed a scale for psychological factors in sports psychology have used the classical test theory (CTT) as their framework. The CTT is a valuable instrument for presenting a typical framework for psychological scale development, based on statistical validation, but it has been criticized regarding the power and reliability of the scale [22].

Since the CTT assumes the ordinal scale to have the exact measurements as a ratio scale or interval scale without objective verification, it induces ambiguity during phenomenological interpretation and inference [23,24], thereby impairing the reliability of the scale. Further, the CTT involves developing a scale based on a criterion of three-dimensional validity of content structure, the criterion itself being based on the validity concept proposed by Cronbach [25]. Hence, researchers’ experiences and subjective judgments, as opposed to objective statistics, are frequently involved in the complete process from developing items and determining the rating scale to identifying a factor structure for the theoretical model. Thus, one critical shortcoming of the CTT is that it undermines the validity of the developed scale, as the reproducibility of the theoretical model is diminished from not using objective indices, such as participants’ levels or item difficulties [22].

To address the difficulties of scale development using the CTT, the Rasch model in item response theory (IRT) [26,27,28,29,30,31] and unified validity testing with a unidimensional validity, confirmed by the 1985 Standards for Educational and Psychological Testing (SEPT) [22,32,33,34,35,36], have been proposed. Accordingly, we aimed to identify SPS for skaters and develop a Sport Psychological Skills Scale (SPSS) for them using the Rasch model of the IRT and a unified framework of validity. The outcomes of this study could provide theoretical references for enhancing skaters’ performances.

## 2. Materials and Methods

### 2.1. Participants

The study population was divided into two groups: skating experts and skaters. Skating experts, with professional knowledge and experience, were selected through purposive sampling. As shown in Table 1, the expert group comprised five researchers (sport psychology professors or PhDs) and four national skating team coaches and athletes.

As shown in Table 2, for the evaluation of the construct validity of SPSS, the population was set to skaters, and skaters who competed in the 2020–2022 skating world championships and Beijing 2022 Winter Olympics were convenience-sampled with the cooperation of the Korea Skating Union (KSU) and International Skating Union (ISU). Data were collected via an online questionnaire. The questionnaire was a self-report survey, and the data were submitted immediately upon answering the questions. A total of 484 questionnaires were collected, and after excluding 28 questionnaires with missing responses, 456 questionnaires were randomly divided into two groups (group A and group B) for construct validity evaluation.

### 2.2. Data Analysis

As shown in Table 3, in this study, we adopted Benson’s [33] construct validity program and Wolf and Smith’s [35] Rasch validity framework, based on the unified validity construct, for which a consensus was reached at the 1985 SEPT. The statistical significance (α) was set at 0.05 for all analyses.

In the substantive domain, Andrich’s [28] Rating Scale Model (RSM), a polytomous Rasch model, was applied to present substantive evidence for establishing a unified validity of the SPSS for skaters, and WINSTEPS 3.65 [37] software (MyCommerce, Chicago, IL, USA) was used. Item fit was assessed based on the mean square fit statistic (MNSQ) and point-biserial correlation coefficient (PBC), and the suitability of the rating scale was determined through rating scale analysis [23,37].

In the structural domain, exploratory structural equation modeling (ESEM), Bayesian structural equation modeling (BSEM), and Maximum Likelihood confirmatory factor analysis (ML-CFA) were applied, as specified, to present the structural domain evidence to establish unified validity of the SPSS for skaters, and Mplus 7.11 [38] software (New York University, New York, NY, USA) was used.

First, we concurrently conducted parallel analysis and ESEM to explore the factor structure of the SPSS for skaters. Parallel analysis was conducted following the protocol proposed by Timmerman and Lorenzo-Seva [39], and ESEM was conducted using maximum likelihood estimation with the Geomin oblique rotation as a rotation method [40]. Moreover, factor coefficients were assessed based on statistical significance and effect size.

ESEM shows the statistical significance of factor coefficients, as suggested by Jennrich [41]. We excluded items with a statistically significant cross-loading at 0.2 or higher when considering interpretability [42]. Through this process, we applied weights to ensure we identified a stable factor structure with minimal cross-loading.

To secure evidence for the factor structure stability of the SPSS for skaters, we employed BSEM and ML-CFA. To address the shortcoming of BSEM, wherein the factor coefficient in a general CFA model is fixed to zero and, thus, does not reflect real-world phenomenon, we used the Markov chain Monte Carlo (MCMC) method to estimate cross-loadings at the 95% interval (−0.2–0.2) [43]. We also performed ML-CFA, because if the factor loadings, other than the cross-loaded factors, are low, constraining them to zero for statistical evaluation is more helpful in follow-up research [44].

In the external domain, we performed latent mean analysis to confirm factorial invariance of the SPSS for skaters across genders.

## 3. Results

### 3.1. Substantive Domain

#### 3.1.1. Classification of SPS

For the theoretical construct of SPS for skaters, the SPS was classified and formulated using Devellis’ [45] theory to provide clarity, a principle for scale development, and Hively’s [46] domain-referenced testing, following existing literature on SPSS and SPS for skaters [15,20,47,48,49,50]. As shown in Table 4, SPS in skaters included image training, fighting spirit, goal setting, emotional regulation, determination, positive self-talk, concentration, and confidence.

A total of 36 items, with four items for each of the nine SPS, were developed, based on the item development method proposed by Crocker and Algina [51], following the criteria for accuracy of content, grammar, conveyance of meaning, and positively worded items. Further, the rating scale was set to the typically used [52] five-point Likert scale, considering that participants are likely to choose “neutral” if they are uncertain whether the corresponding SPS was used.

#### 3.1.2. Content Validity

To ensure an objective content validity evaluation, content validity was tested following Aiken [53] and binomial probability distribution. An expert panel was asked to rate each item on a scale from one to five to determine whether the item described the corresponding SPS effectively and was appropriate for skaters. The total score for each item (S) was computed by dividing the sum of all scores by the product of the number of raters (N) and the difference of maximum (high) score and minimum score (low). Subsequently, the significance of the V statistic was assessed following the binomial probability distribution for each item.

A V statistic close to 1 and significance of below 0.05 suggested that the corresponding item contained valid content for the SPSS for skaters. Conversely, a V statistic close to 0 and a significance of 0.05 or higher suggested that the corresponding item did not contain valid content for the SPSS for skaters. As shown in Table 5, 9 out of 36 items (items 4, 5, 6, 7, 8, 9, 10, 14, 21) had a V statistic close to 0 and a significance of 0.05 or higher. In other words, these items were not appropriate for the SPSS for skaters and, thus, were removed from the item pool, and the 27 remaining items were used for the SPSS for skaters.

#### 3.1.3. Test of Uni-Dimensionality

Before validating the factor structure, the unidimensionality of the factor structure, a prerequisite of the Rasch model, was tested. Principal component analysis of unidimensionality was analyzed using WINSTEPS 3.65 [37] software, and, as shown in Table 6, the explained variance was 39.6%, meeting the condition for unidimensionality (observed variance ≥ 20%) [54]. Hence, the items on the SPSS for skaters were confirmed to be unidimensional.

#### 3.1.4. Appropriateness of Rating Scale

Rating scale analysis of the Rasch model was performed to evaluate the suitability of the rating scale. Suitability was evaluated based on probability curves and the following criteria [37,54]: (1) at least ten counts per category, (2) frequency distribution across categories, (3) proportional relationship between category and average measure (AM) of the category, (4) satisfaction of the standardized infit and outfit criteria for each category, and (5) change of step calibration (SC). There was an even distribution of counts across categories, and frequency and AM monotonically increased across categories. Further, the infit and outfit values for each category were within 7.5–1.30. However, the absolute value of SC between category 2 and category 3 was smaller than 1.4, failing to meet the criterion (1.4–5.0). Therefore, a five-point rating scale was inappropriate for the SPSS for skaters.

To identify the optimal rating scale, we tested different rating scales. As shown in Table 7, there were at least ten counts per category with a four-point Likert scale, and the counts were evenly distributed across categories. The AM monotonically increased across categories. The infit and outfit values for each category were within 7.5–1.30 [55], and the absolute value of SC was also within 1.4–5.0. Therefore, a four-point Likert scale was appropriate for the SPSS for skaters.

#### 3.1.5. Item Relevance

Relevance of the items on the SPSS for skaters was evaluated using the Rasch rating scale model [28]. Item relevance refers to the discriminatory power of each item, and MNSQ (mean square fit statistic) and PBC (point-biserial correlation) were used to determine relevance.

The MNSQ criteria proposed by Hong [54] and McNamara [55] (0.75–1.3), and the PBC criteria proposed by Wolfe and Smith [35] (≥0.30), were used. As shown in Table 8, the MNSQs (infits or outfits) for items 16, 23, 24, 26, 29, 32, and 36 were greater than 1.3. This suggested that these items had less relevance to SPS in skaters or were redundant with other items and, thus, did not accurately distinguish between different SPSs. Therefore, these items were removed from the item pool for the SPSS for skaters.

### 3.2. Structural Domain

#### 3.2.1. Parallel Analysis (PA)

To explore the factor structure of the SPSS for skaters, PA was performed first, to infer the number of factors. As shown in Table 9, the real-data eigenvalue was smaller than the 95% of random eigenvalue from the six-factor model. Thus, five or fewer factors were inferred suitable for the SPSS for skaters [39].

#### 3.2.2. Exploratory Structural Equation Modeling

As the number of factors was inferred to be five or fewer in the PA, we compared the fit of three-factor to seven-factor models [56], and a model with a factor structure with relatively good fit indices and interpretability was selected as the final model.

As shown in Table 10, the comparative fit index (CFI) and Tucker Lewis index (TLI) were 0.90 or higher, and the root mean square error of approximation (RMSEA) was 0.80 or lower from the four-factor model. Thus, the factor structure was determined from four factors and by considering the factor coefficients and interpretability [54,57,58].

As shown in Table 11, in the ESEM, items with multidimensionality, with a cross-loading of 0.2 or higher for at least two factors [42], and items that had been loaded entirely differently from the theoretical model and hindered practical interpretation [45], were observed, and these items were eliminated through repeated estimation. Due to this process, four items were deleted, finalizing the SPSS for skaters to a five-factor, 17-item structure. Moreover, the correlations among the factors were below 0.80, satisfying Kline’s [59] criterion for discriminant validity.

The factors were named following the contents of their items. Factor 1 contained the following items: “not giving up despite being physically drained”, “not giving up even while losing the game”, and “feeling more energy when competing against a rival or rival team”, so factor 1 was named “Fighting spirit”.

Factor 2 comprised the following items: “regulating anxiety”, “not easily stirred”, and “remaining calm during the game, even amid a setback”, and was named “emotional regulation”.

Factor 3 comprised the following items: “engaging in encouraging self-talk”, “telling oneself that everything will be fine during a setback”, and “cheering oneself up when facing challenges”, was named “positive self-talk”.

Factor 4 comprised the following items: “concentration unaffected by situations” and “not losing concentration despite sabotage”, and was named “concentration”.

Factor 5 comprised the following items: “frequently imagining success”, “frequently imaging challenging technical skills mentally”, and “imaging different technical skills depending on the competitor”, and was named “image training”.

#### 3.2.3. Bayesian Structural Equation Modeling

As shown in Table 12, BSEM was performed for cross-validation of the SPSS factor structure extracted from ESEM.

As shown in Table 13, the standardized coefficients of the items comprising the five factors (fighting spirit, emotional regulation, positive self-talk, concentration, and image training) were statistically significant at above 0.500 (*p* ˂ 0.001). Moreover, none of the items had cross-loadings, confirming that the SPSS for skaters had a stable factor structure.

ML CFA (maximum likelihood Confirmatory Factor Analysis).

CFA using ML estimation was performed to evaluate the factor structure fit for the SPSS for skaters.

#### 3.2.4. Reliability and Difficulty by Factors

The reliability of the responses and items of the SPSS for skaters was analyzed. As shown in Table 14, the separation index was greater than 2.0, with a reliability of 0.80 or higher, and infit and outfit of 0.75 or higher to less than 1.3 [54], confirming good reliability of items and responses. These results showed that the SPSS for skaters was suitable for measuring SPS in skaters, and it could measure skaters’ SPS accurately.

### 3.3. External Domain

Multigroup CFA (MCFA)

In the external domain, multigroup CFA (MCFA) was performed to examine the power of the SPSS for skaters.

Measurement invariance of the factor structure was assessed through MCFA. As shown in Table 15, the difference between the unconstrained model and the metric equivalence model (∆χ^2^), at 95% confidence level, was 0.137, and, thus, was not significant (>0.05). These results suggested that the SPSS for skaters had measurement invariance across gender, indicating its validity and reliability as a scale for measuring SPSS in skaters.

## 4. Discussion

This study aimed to identify SPS currently employed by skaters in a competitive setting and to develop a scale for measuring their SPS.

We classified SPS among skaters based on existing literature and a domain-referenced approach, and SPS in skaters were categorized as image training, fighting spirit, goal setting, emotional regulation, determination, positive self-talk, concentration, emotion, and confidence. Subsequently, items were developed following the method suggested by Crocker and Algina [51]. The developed items were evaluated for their relevance and correlation by experts, and the content validity of the items was tested using Aiken’s [53] V coefficients. Through this process, 27 out of 36 items were chosen for the SPSS for skaters, and these items were evaluated for construct validity to develop the SPSS for skaters.

Studies developing scales in sports have primarily evaluated content validity based on researchers’ subjective judgments. However, such an approach can diminish the stability of the theoretical model of the scale or undermine its reliability [60]. Hence, subsequent studies that attempt to develop a psychological scale in sports are recommended to develop items using Aiken’s [53] V coefficients.

Construct validity was assessed following unified validity testing, using the Rasch model, and the factor structure of the SPSS for skaters was explored through methods with optimal factor structure [22,42,57], including PA, ESEM, BSEM, and ML-CFA.

One benefit of ESEM is that the interpretability of a factor structure can be easily assessed based on the significance of item factor loadings and effect size. In the present study, we assessed the factor loadings based on the statistical significance criteria proposed by Jennrich [41], and items with a statistically significant factor loading at 0.20 or higher were deleted. Hence, a five-factor, 17-item structure was extracted as the optimal factor structure for the SPSS for skaters. We also assessed the stability and fit of the factor structure through BSEM and ML-CFA, and, based on the results, the SPSS for skaters was finalized to a five-factor, 17-item structure. The reliability of the SPSS for skaters was assessed at two dimensions (item and response), and the results confirmed that the SPSS for skaters was appropriately targeted, and item reliability was also excellent at above 0.80. Finally, the SPSS for skaters was confirmed equally valid for both genders.

Through this procedure, skaters’ SPS were identified as fighting spirit, emotional regulation, positive self-talk, concentration, and image training. Most existing studies on SPS in skaters introduced anxiety, depression, fighting spirit, confidence, goal setting, imagery, team harmony, and willpower as the primary SPS among skaters, but positive self-talk, concentration, and image training were not included. Such discrepancy of findings may be due to researchers’ subjective judgment and unquestioning application of SPS scales developed for athletes in other types of sports, as in existing studies on SPS in skaters, as opposed to the difference in the study population.

Martens [21] mentioned the need to develop and implement a sport-specific psychological skill scale tailored to the characteristics of athletes in a particular sport, rather than using a scale universally used across all sports, to measure athletes’ psychological factors empirically. This is because psychological factors are highly detailed and unique, and, thus. using the same approach as that of a non-psychological scale limits the measurement of psychological factors in athletes.

Skating is a highly competitive and relative sport compared to other sports. Winners are determined based on records and through races. Thus, athletes undergo dramatic changes in emotions during the competition and are bound to be heavily influenced by their psychological states [17,20]. Further, cornering skills and physical fitness are closely linked to game outcomes; hence, athletes undergo skills training for cornering [18]. Considering this, positive self-talk, concentration, and image training, identified as SPS factors in this study, could be considered valid SPS to be utilized by skaters during active control of emotions and skills learning.

SPS employed by skaters were identified to be fighting spirit, emotional regulation, positive self-talk, concentration, and image training; hence, SPS training programs focused on these SPS should be developed and implemented as strategies to enhance skaters’ performances.

Although we identified the SPS employed by skaters, our findings did not explain the roles of SPS in athletes’ overall athletic performance. Hence, subsequent studies should investigate the values and roles of SPS in skaters.

## 5. Conclusions

This study aimed to explore SPS used by skaters and to develop a scale for measuring SPS in skaters, and the following conclusions were drawn. First, SPS employed by skaters were fighting spirit, emotional regulation, positive self-talk, concentration, and image training. Second, the SPSS for skaters was finalized to five factors and 17 items using a four-point Likert scale, and the scale was confirmed to have good reliability and validity.

## Figures and Tables

**Table 1 ijerph-19-08035-t001:** General characteristics of the expert group.

Group		Sex	N
Experts	Professor in sport psychology	Male	1
		Female	1
	doctor in sport psychology	Male	1
		Female	2
	Coach	Male	2
		Female	2

**Table 2 ijerph-19-08035-t002:** General Characteristics of Survey Participants.

Dimain			N	%	Dimain			N	%
Group A (228)	sex	Male	172	75.4	Group B (228)	sex	Male	168	73.6
		Female	56	24.6			Female	60	26.4
	career	Less than 4 yrs	21	9.2		career	Less than 4 yrs	35	14.0
		5 to 7 yrs	57	25.0			5 to 7 yrs	68	29.8
		8 to 10 yrs	105	46.0			8 to 10 yrs	101	44.3
		More than 11 yrs	45	19.7			More than 11 yrs	24	10.5
	event	speed	130	57.0		event	speed	135	59.2
		short track	98	43.0			short track	93	40.8

**Table 3 ijerph-19-08035-t003:** Construct Validity Analysis.

Domain	Analysis Method	Data Source
Substantive	• Item development according to theoretical model • Content validity verification	Experts
	• Rasch model: Unidimensionality verification/Conformity verification	Group A
	• Rasch model: Response category verification	
Structural	• Parallel analysis, Exploratory Structural Equation Modeling, Bayesian Structural Equation Modeling	
	• Maximum Likelihood CFA, Reliability analysis	Group B
External	• Multi-Group Confirmatory Factor Analysisis	

**Table 4 ijerph-19-08035-t004:** Speed Skater’ Eight Key Sport Psychological Skills.

Factor	Definition
Imagery training	Imagining successful scenarios or important behavioral skills and movements.
Fighting spirit	Willingness and behavior to do one’s best to the end
Goal setting	Setting specific goals for training and competition.
Emotion regulation	Skill to control the intensity of negative emotions, or convert them into positive emotions.
Will	Firm commitment to achieve a goal or to win.
Positive Self-talk	Skill to overcome difficulties and encourage oneself through self-talk
Concentration	Skill to focus on training and competition, regardless of circumstances.
Confidence	Positive belief in one’s own abilities and performance.

**Table 5 ijerph-19-08035-t005:** Items Assessing Speed Skaters’ Sport Psychological Skills.

Item	V Coefficient	Item	V Coefficient	Item	V Coefficient
4	0.10	8	0.15	14	0.08
5	0.12	9	0.19	21	0.18
6	0.08	10	0.14		

**Table 6 ijerph-19-08035-t006:** Verification of Unidimensionality.

		Eigenvalue	%
Explained variance	Person	22.5	27.0
	Item	17.1	20.5
Unexplained variance		43.7	52.4
Total variance		83.3	100

**Table 7 ijerph-19-08035-t007:** Verification of the 4-point Likert Scale’s Appropriateness.

Category	Count	%	AM	Infit	Oufit	SC
1	2078	12	−0.30	1.17	1.16	
2	6119	34	−0.18	0.88	0.90	−1.89
3	7117	41	0.63	0.91	0.89	−0.18
4	2523	14	1.78	1.28	1.01	2.07

**Table 8 ijerph-19-08035-t008:** Results of Item Relevance Verification.

Item	MNSQ		PBC	Item	MNSQ		PBC
	Infit	Outfit			Infit	Outfit	
16	1.29	1.40	0.20	29	0.70	0.68	0.49
23	1.59	1.62	0.40	32	1.40	1.42	0.57
24	1.88	1.84	0.50	36	1.50	1.49	0.42
26	1.68	1.62	0.51				

**Table 9 ijerph-19-08035-t009:** Parallel Analysis Results.

Factor	Real-Data Eigenvalues	95% of Random Eigenvalues	Comparison
1	19.221	1.557	Real-data > 95% random
2	3.882	1.501	Real-data > 95% random
3	2.595	1.477	Real-data > 95% random
4	1.823	1.451	Real-data > 95% random
5	1.574	1.403	Real-data > 95% random
6	1.307	1.388	Real-data > 95% random

**Table 10 ijerph-19-08035-t010:** Comparison of Factors’ Fit.

Number of Factors	χ^2^	df	CFI	TLI	RMSEA	RMSEA 90% CI
3	6883.41	1921	0.878	0.880	0.093	0.088–0.094
4	6003.22	1899	0.900	0.901	0.072	0.080–0.088
5	4998.32	1724	0.906	0.900	0.069	0.065–0.072
6	4322.10	1698	0.910	0.904	0.061	0.059–0.068
7	4021.23	1475	0.911	0.905	0.059	0.050–0.067

**Table 11 ijerph-19-08035-t011:** Results of ESEM Analysis.

	Items	Fighting Spirit	Emotional Regulation	Positive Self-Talk	Concentration	Image Training
3	Even if your stamina is exhausted, you do not give up until te end.	0.584 *	−0.060	−0.005	0.043	0.008
2	I don’t give up even when the game gets bad.	0.708 *	0.028	−0.007	−0.007	0.047
1	When I play against a competitor or team, my strength rises.	0.719 *	−0.049	0.096	0.013	0.008
28	When I get anxious during a game, I control my anxiety through deep breathing.	0.109	0.591 *	0.093	0.011	0.104
31	I don’t get excited easily during the game.	0.105	0.474 *	0.017	−0.043	0.009
30	I play games calmly even in a crisis situation.	−0.044	0.783 *	−0.046	0.017	0.010
12	I always have encouraging conversations with myself.	−0.100	0.055	0.756 *	0.099	0.084
13	In every difficult moment, I tell myself that everything will be fine.	−0.049	0.211	0.702 *	0.098	−0.025
11	Whenever I am in trouble, I tell myself to be strong.	0.077	0.192	0.604 *	−0.093	−0.045
15	When I lose a match, I encourage myself by talking to myself.	−0.010	0.204	0.866 *	0.055	0.131
34	I do not lose concentration regardless of the circumstances around me.	−0.004	0.082	−0.199	0.539 *	0.079
33	I do not lose concentration even in difficult situations.	−0.003	0.009	−0.023	0.716 *	−0.042
35	I am not distracted by the disturbances around me.	0.049	0.010	−0.108	0.706 *	0.003
18	I always imagine a scene when a skill is succeeding.	0.053	0.069	0.032	−0.022	0.677 *
17	I often imagine highly technical scenes.	0.067	0.045	0.038	0.014	0.805 *
19	I imagine important things about operations or technical moves.	−0.020	−0.031	0.018	0.024	0.707 *
20	Draw technical moves according to the type of opponent.	0.049	0.068	0.017	0.093	0.731 *
		**Fighting Spirit**	**Emotional Regulation**	**Positive Self-Talk**	**Concentration**	**Image Training**
emotional regulation		0.439	1			
positive self-talk		0.332	0.302	1		
concentration		0.324	0.340	0.362	1	
image training		0.282	0.403	0.353	0.403	1

* *p* < 0.05.

**Table 12 ijerph-19-08035-t012:** Speed Skater Sports Psychological Skills Scale Factor Structure Applying BSEM.

	Fighting Spirit	Emotional Regulation	Positive Self-Talk	Concentration	Image Training
3	0.599 ***	0.060	0.007	0.072	0.008
2	0.771 ***	−0.074	0.013	0.009	0.047
1	0.734 ***	0.033	0.030	0.008	0.008
28	−0.004	0.564 ***	−0.014	0.031	0.104
31	0.016	0.692 ***	0.051	−0.024	0.009
30	0.030	0.776 ***	−0.077	0.017	0.010
12	0.040	0.059	0.626 ***	−0.074	0.084
13	0.004	0.119	0.722 ***	0.019	−0.025
11	0.014	−0.030	0.881 ***	−0.010	−0.045
15	0.022	−0.005	0.798 ***	0.043	0.131
34	0.003	−0.036	−0.074	0.772 ***	0.079
33	−0.037	0.009	0.019	0.708 ***	−0.042
35	0.049	0.016	−0.010	0.699 ***	0.003
18	0.011	−0.036	0.016	−0.074	0.705 ***
17	0.019	−0.050	−0.005	0.019	0.739 ***
19	−0.020	0.020	0.019	−0.014	0.665 ***
20	−0.049	0.022	0.013	−0.007	0.720 ***

*** *p* < 0.001.

**Table 13 ijerph-19-08035-t013:** Results of ML CFA.

Factor	Item	Standardized Corfficient	Standard Error	*Z* Value	*p* Value	R^2^
fighting spirit	3	0.809	0.018	44.93	<0.001	0.654
	2	0.750	0.013	57.67	<0.001	0.562
	1	0.635	0.016	32.57	<0.001	0.403
emotional regulation	28	0.638	0.018	34.50	<0.001	0.407
	31	0.712	0.016	44.47	<0.001	0.507
	30	0.729	0.019	38.35	<0.001	0.531
positive self-talk	12	0.746	0.018	41.38	<0.001	0.556
	13	0.758	0.015	50.51	<0.001	0.574
	11	0.722	0.016	45.13	<0.001	0.521
	15	0.765	0.016	47.80	<0.001	0.585
concentration	34	0.716	0.017	42.10	<0.001	0.512
	33	0.688	0.018	38.21	<0.001	0.473
	35	0.693	0.022	31.45	<0.001	0.480
image training	18	0.726	0.019	38.20	<0.001	0.527
	17	0.713	0.017	41.93	<0.001	0.508
	19	0.746	0.017	43.81	<0.001	0.556
	20	0.776	0.018	43.10	<0.001	0.602
χ^2^ = 6670.43, df = 309, CFI = 0.901, RMSEA = 0.057						

**Table 14 ijerph-19-08035-t014:** Response and Item Reliability.

	Factor	SEP	Rel.	Infit	Outfit
RR	Fighting spirit	3.99	0.91	1.00	1.01
	emotional regulation	3.87	0.90	1.03	1.02
	positive self-talk	3.91	0.91	0.99	1.00
	concentration	4.01	0.93	0.96	0.97
	image training	4.11	0.94	10.01	1.00
IR	Fighting spirit	4.00	0.95	1.00	1.00
	emotional regulation	3.98	0.90	0.99	1.00
	positive self-talk	4.02	0.93	1.03	1.00
	concentration	3.98	0.90	0.98	0.99
	image training	4.02	0.91	1.00	1.01

**Table 15 ijerph-19-08035-t015:** Results of Verification of Measurement Equivalence by Gender.

	χ^2^	∆χ^2^	df	*p*	∆df	RMSEA
Unconstrained model	823.12		307			0.045
Factor coefficient same constraint	849.21	26.09	344	0.137	37	0.049
Covariance equal constraint	899.21	76.09	360	0.157	53	0.058
Factor coefficient/Covariance/ Error variance	911.83	88.71	381	0.140	74	0.069

## Data Availability

The data presented in this study are available on request from the corresponding author. The data are not publicly available due to privacy issues.

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
