# Peer review of "Exploring and Developing a Scale Using Item Response Theory for Sport Psychological Skills in Speed Skaters"

_ijerph, 2022, doi:10.3390/ijerph19138035_

Round 1
Reviewer 1 Report
The purpose of the study was to elaborate the psychology skill scale for skaters. But the results have not been clearly presented. The presence of table 5 dedicated to taekwondo athletes is not entirely appropriate. In addition, the tables are too cluttered and low the perception of the results.
Author Response
It was revised in response to the reviewers' comments and suggestions.Reviewer 2 Report
The article is very interesting and can have interesting applications for the future. The part of statistical analysis is very complete and specific, although a bit long to follow.
However, there are things to improve regarding the form.
The abstract in my opinion should be rewritten. There is no reference to why the study was conducted, and it is unclear because it has too many abbreviations. In my opinion, it would be more correct to indicate with the full name the analyses carrid out. The abtract should capture the reader’s interest in bringing him to read the entire article, instead this abstract does not.
In the Introduction part, there are two main problems: the lack of references and the division in paragraphs. Many parts of the introduction seem to lack references, such as lines 25-27 or even lines 54-61. It would be better to add them to get a clearer idea of the article’s sources. In addition, the division of paragraphs seems to be done with little criterion. For example, lines 28-47 should be a single paragraph, as also lines 54-61. In the way they are currently presented, they seem to be unrelated, but they are not.
Materials and methods. Line 98 is not so clear. I think I have understood that skating expert have been asked to create the assessment scale, but I am not sure, because the part is not clear. It should be rewritten and explained better. In addition, I think that Table 1 does not add much more information, in my opinion it could be removed. Also, because in the article there are really many tables.
In line 104 there is a mistake in the year (20020). In table 3, notes explaining abbreviations within the table are missing. It is not clear to me why different software was used for different types of analysis. Was it not possible to use just one?
Results. Table 5 is not so clear. Is it related to the above analysis or is it from another article about Taekwondo practitioners? Lines 185-186 are a repetion of the methods, it should be removed. Lines 194-202 are describing the methods of analysis, Perhaps they should be reported in that section more than in the results part. The tables as a whole are a bit too many. Perhaps it would be worth considering if some can be inserted directly as an appendix file.
Even the discussion part has the same problem as the introduction part with the division in paragraphs.
Author Response
We thank the reviewers for their careful review. We have accepted criticisms and suggestions, and have modified them as much as possible. thank you
Reviewer 3 Report
Review of “Exploring and developing a scale using item response theory for sport psychological skills in speed skaters”
The paper is well written and methodologically and theoretically thorough. The abstract, however, needs to be revised, as it lacks precision and it is what the public first reads. I think the paper should be accepted after minor revisions will be made by the authors.
Major revisions:
Abstract
Lines 16-17. These sentences are unclear. Maybe the authors could write “and TLI and RMSEA were used as fit indices for ESEM, BSEM, ML CFA, and MCFA”. I would definitely remove reliability, as, to my knowledge, these are indices of validity, and not reliability.
Lines 18-19. At the moment, I do not understand why the authors are talking about transformative leadership here, maybe I will understand reading the remaining part of the document, but it should already be clear in the abstract.
Introduction
Lines 24-27. In the first paragraph, the first sentence and the second one have the same content, they are just rephrased. Please remove one of them.
Lines 64-66. This sentence is unclear, and anxiety and depression are not psychological skills, maybe the ability to manage them is a skill. Moreover, I am not sure if it is possible to consider a skill a team attribute, to my understanding skills are individual, so it is difficult to say ‘team harmony’ is a skill, I would rather say that the ability to develop positive social relationships is a skill. I think the authors could write: “emotion regulation, fighting spirit, confidence, goal setting, imagery, interpersonal skills, and determination are all required skills for the enhancement of skating performance. However, most studies […]”.
Lines 89-95: I think scale development guidelines adopted for this study are overtaken. Guidelines from 1985 have been quoted, however we have much more recent guidelines for scale development, and I would challenge authors to provide me more recent references.
Results and Discussion
Table 11: The item 31, “I don’t get excited easily during the game”, can be misinterpreted as “I get bored during the game” and, in facts, it has a poor loading in the ESEM. Have the authors considered to propose its rephrasing in future studies?
Minor revisions:
Abstract
Line 10: maybe “sport psychology skills scale for skaters”?
Introduction
Line 65: “imagery”?
Materials and Methods
Table 1: “Doctor in sport psychology”
Line 104: “2020-2022”
Table 3: “Reliability analysis”
Line 126: Please, spell out ML in ML-CFA
Line 151: “DeVellis”
Author Response

(The authors gave the same response as above.)

Round 2
Reviewer 2 Report
I thank the authors for having fixed and corrected the article, now it is ready for publication